# Epigenetic Regulatory Processes Involved in the Establishment and Maintenance of Skin Homeostasis—The Role of Microbiota

**DOI:** 10.3390/ijms26020438

**Published:** 2025-01-07

**Authors:** Kornélia Szabó, Fanni Balogh, Dóra Romhányi, Lilla Erdei, Blanka Toldi, Rolland Gyulai, Lajos Kemény, Gergely Groma

**Affiliations:** 1HUN-REN-SZTE Dermatological Research Group, 6720 Szeged, Hungarykemeny.lajos@med.u-szeged.hu (L.K.); groma.gergo@gmail.com (G.G.); 2Department of Dermatology and Allergology, Albert Szent-Györgyi Medical School, University of Szeged, 6720 Szeged, Hungary; 3HCEMM-USZ Skin Research Group, 6720 Szeged, Hungary

**Keywords:** epigenetics, skin

## Abstract

Epigenetic mechanisms are central to the regulation of all biological processes. This manuscript reviews the current understanding of diverse epigenetic modifications and their role in the establishment and maintenance of normal skin functions. In healthy skin, these mechanisms allow for the precise control of gene expression, facilitating the dynamic balance between cell proliferation and differentiation necessary for effective barrier function. Furthermore, as the skin ages, alterations in epigenetic marks can lead to impaired regenerative capacity and increased susceptibility to environmental stressors. The interaction between skin microbiota and epigenetic regulation will also be explored, highlighting how microbial communities can influence skin health by modulating the host gene expression. Future research should focus on the development of targeted interventions to promote skin development, resilience, and longevity, even in an ever-changing environment. This underscores the need for integrative approaches to study these complex regulatory networks.

## 1. Introduction

In recent decades, research has demonstrated that environmental exposures often trigger dynamic, sometimes heritable changes in our bodies, playing a crucial role in our ability to rapidly adapt to changing environmental factors. This is particularly significant in today’s accelerated world, where, especially since the Industrial Revolution, changes in our living conditions, diet, and hygiene have occurred more quickly and perhaps more extensively than at any other time in history. These types of changes are known as epigenetic changes. They may or may not be inherited and cause stable cellular alterations without changing the DNA sequence, thereby influencing the transcriptional potential of cells [1,2]. Since its discovery, it has become apparent that epigenetic regulation allows for a more modular and rapid form of adaptation than evolutionary processes and the inherited changes that occur through mutations. In recent years, researchers have identified numerous types of epigenetic regulatory processes, with DNA methylation, DNA hydroxymethylation, covalent histone modifications, ATP-dependent chromatin accessibility, major chromatin architectural changes, three-dimensional regulatory interactions, non-coding RNAs, and RNA methylation being the most recognized and extensively studied. Although this list is extensive, it is likely not exhaustive. Nonetheless, it clearly illustrates the broad spectrum of mechanisms through which the regulation of environmental adaptation may be achieved in the body [3].

Maintaining body and skin homeostasis is a highly complex process that requires the continuous regulation of numerous intricate functions. This includes the maintenance of the skin’s complex barrier, which must remain effective despite constantly evolving external and internal environmental factors. To ensure the skin’s ability to adapt to these changes, a robust and precise epigenetic regulatory mechanism is essential. This mechanism is critical for healthy differentiation and in the later stages of aging [4]. While this adaptability is beneficial and contributes to our plasticity, it is evident that changes caused by adverse environmental influences can sometimes also disrupt and upset the balance of the organism, leading to the development of various pathological conditions [5], among which we can find chronic inflammatory diseases (e.g., type 2 diabetes, cardiovascular diseases, asthma, allergies) and various types of cancer.

The objective of this article is to summarize our current understanding of epigenetic processes in the skin under healthy conditions, including the development of the skin and the aging process. Furthermore, the impact of microbiota on the maintenance of cellular homeostasis is examined.

## 2. Epigenetic Regulation

Epigenetic mechanisms play a pivotal role in determining gene expression and functional outcomes, thereby introducing additional layers of complexity to cellular processes. These mechanisms often act independently, whereby alterations in the pattern of a specific mark at a specific locus can already result in changes to gene expression and function. However, this picture is further complicated by the fact that their combined pattern can also be dominant, sometimes collectively shaping the resulting epigenome in response to a given effect, greatly increasing the complexity of the processes [6]. This chapter provides an overview of the currently known elements of the epigenetic regulatory system.

### 2.1. DNA-Associated Epigenetic Regulation: Methylation and Demethylation

One of the most extensively studied epigenetic modifications is DNA methylation, which plays a crucial role in regulating gene expression. This process involves the covalent attachment of a methyl (CH3) group to the C5 position of cytosine residues in a dinucleotide (CC) or oligonucleotide (CGGCGG) context within the so-called CpG islands. This results in the formation of a 5-methylcytosine (5mC) mark. These methyl groups extend into the major groove of DNA, altering its biophysical properties [7]. This mark primarily promotes and stabilizes gene silencing in a multitude of ways (Figure 1) [8]. The removal of the 5mC content of the genome produces 5-hydroxymethylcytosine (5hmC), 5-formylcytosine (5fC), and 5-carboxylcytosine (5caC) [9,10]. The specific role of these intermediates in epigenetic regulation remains unclear, and it is even possible that they are simply intermediates during DNA demethylation [11]. Nevertheless, further investigation should address these issues.

The dynamics of DNA methylation are controlled by DNA methyltransferases (e.g., DNMT1, DNMT3A, and B) and demethylases (e.g., TET1, 2, 3) [12]. DNA methyltransferases transfer a methyl group from S-adenylmethionine to the fifth carbon of cytosine. De novo methylation adds methyl groups to previously unmethylated CpG sites (DNMT3A and B), maintains methylation, preserves methylation patterns after DNA replication (DNMT1), and there is even a replication-independent form of the latter (DNMT1) [13,14,15]. Demethylation can be passive, occurring due to a lack of DNMT1 during replication, or it can be active, involving the enzyme-mediated removal of the methyl group from 5-methylcytosine [16]. Generally, promoter methylation represses gene expression, while hypermethylation in the gene body is linked to gene activation [17].

### 2.2. Epigenetic Regulation of Histones

The nucleosome, the fundamental unit of chromatin, is composed of 147 base pairs of genomic DNA wrapped around a core histone octamer, which consists of an octamer of two molecules of H2A, H2B, H3, and H4 [18]. Epigenetic regulation at the histone level occurs in three stages (Figure 2) [19]. (1) Histone chaperones facilitate the transport, incorporation, and exchange of histones into chromatin; (2) the histone composition and its variants are controlled by these chaperones. Furthermore, (3) a multitude of histone post-translational modifications [20,21] regulate the chromatin structure and gene expression in conjunction with DNA epigenetic modifications [19,22].

#### 2.2.1. Histone Chaperones

Histone chaperones protect histones from non-specific interactions and aggregation during nuclear transport [23], ensuring their proper incorporation into chromatin [24]. They regulate histone metabolism by controlling histone assembly and disassembly on DNA [25,26,27], and by facilitating the exchange of histone variants [28], they influence chromatin dynamics and landscape.

#### 2.2.2. Histone Variants

Histones can be categorized into canonical and variant types, including non-allelic isoforms [29,30]. Canonical histones, or replication-dependent histones, are expressed in a cell cycle-dependent manner [30], are clustered in chromatin [31,32] and have highly similar isoform sequences [33] Their mRNAs lack introns and feature a stem-loop motif at the 3′ end, facilitating rapid synthesis [34]. In contrast, replication-independent histones are expressed throughout the cell cycle [35], exhibit significant sequence diversity, and are associated with specific loci. These histones’ mRNAs contain introns and polyA tails and undergo standard mRNA processing [20,34,36,37].

#### 2.2.3. Post-Translational Modifications of Histones

The core histones are globular proteins with protruding flexible ‘tails’ that can undergo a variety of post-translational modifications (e.g., methylation, acetylation, phosphorylation, ubiquitylation, citrullination, and succinylation) [20,21].

Histone acetylation, the addition of an acetyl group to lysine residues on core histones, generally relaxes chromatin and activates gene expression, while histone deacetylation represses genes [38]. Histone phosphorylation and dephosphorylation, targeting serine, threonine, and tyrosine residues [39], introduce negative charges that loosen chromatin and enhance transcription [40,41], mediated by kinases and phosphatases, respectively [39]. The methylation of histones, particularly on arginine and lysine residues, by methyltransferases and demethylases affects transcription depending on the specific site and histone involved (Figure 3) [42,43]. Histone ubiquitination, notably polyubiquitination, leads to protein degradation [44], while monoubiquitination can either activate or repress transcription depending on the genomic context. For instance, H2A monoubiquitination is linked to repression in satellite regions [45,46], while H2B monoubiquitination is associated with active transcription in gene bodies [47,48].

#### 2.2.4. Crosstalk Among Different Epigenetic Regulatory Mechanisms

The epigenome consists of distinct yet interconnected layers, encompassing the overall structure and organization of chromatin and specific modifications to histones and DNA. Histone variants have unique sequences that can be differentially modified by post-translational modifications (PTMs) [52]. When histone variants are incorporated into nucleosomes by chaperones, they can be decorated with specific PTMs that define the chromatin state, thereby influencing the transition between active and repressed chromatin forms [53,54]. These modifications collectively form a specific pattern known as the “histone code”, which guides chromatin–DNA interactions and determines the functional state of a given locus [55]. The histone code is dynamic, regulated by a variety of proteins—writers, readers, and erasers—that contribute to a highly modular and rapidly changing system, enabling cells to swiftly adapt to environmental and developmental cues [56,57].

Histone chaperones are crucial not only for the deposition of histones and their variants but also for influencing their modification states [58]. Certain chaperones interact with specific histone-modifying enzymes, thereby facilitating targeted modifications of histones during their incorporation into chromatin [59]. Additionally, the presence of post-translational modifications (PTMs) can impact the activity of histone chaperones and the deposition of histone variants [58]. Conversely, the replacement of histone variants by chaperones can modify the PTM landscape, thus altering the chromatin’s functional state [60].

The PTMs of histones are intricately linked to DNA-associated epigenetic regulatory mechanisms, working in a highly coordinated manner. For instance, histone acetylation is known to promote DNA demethylation [61]. However, the effects of histone methylation are context-dependent; recent studies suggest that while H3K4me3, H3K4me1, and H3K27me3 generally inhibit DNA demethylation, H3K9me3 and H3K36me3 are positively correlated with this process. This close interplay is partly mediated by DNA methyltransferases and associated proteins that recognize specific histone PTMs, such as methylation and ubiquitination, through direct interactions [62].

### 2.3. Non-Coding RNAs

The primary function of epigenetic modifications is to regulate the accessibility of chromatin to the transcriptional machinery, thereby controlling the expression rates of individual genetic loci. This regulation is achieved primarily through a complex network of covalent and non-covalent modifications to DNA and histones [63]. While the complexity of this network is remarkable, it represents only a part of the whole picture. Over the past two decades, research has increasingly uncovered the role of additional elements in epigenetic regulation, in particular various non-coding RNAs (ncRNAs). A large number of ncRNAs with different structures and effects have been identified (Figure 4), which can influence epigenetic regulation through a variety of mechanisms.

These mechanisms include modulating the levels of factors involved in these processes or regulating chromatin’s structure and accessibility, targeting specific complexes to defined genomic regions via their scaffolding functions, and regulating transcription and epigenetic memory, among many others [63,64,65]. Not surprisingly, they play an important role in regulating the formation, maintenance, and repair of healthy structure and skin functions, although our knowledge is currently incomplete.

## 3. Epigenetic Regulation in the Skin Under Healthy Conditions

The skin is one of the most exposed organs in the human body, and the formation of the stratified epidermis is a highly complex and continuous phenomenon. Natural differentiation processes, which are complex, stepwise differentiation programs, already result in a high rate of cell turnover. This necessitates a constant equilibrium between cell renewal, differentiation, and cell loss. Moreover, the turnover rate is susceptible to alterations in response to injuries, such as wounding and barrier damage, and is also dependent on the age of the individual [66,67]. This necessitates the presence of a highly modular and efficient system for maintaining tissue homeostasis, which requires the coordinated functioning of all levels of the epigenetic regulatory system (reviewed by Wagner et al. [67]).

### 3.1. Skin Development

Undifferentiated keratinocytes in the interfollicular epidermis originate from mitotically active cells in the basal layer of the epidermis, where two distinct cell subpopulations, namely keratinocyte stem cells and transit amplifying cells, can be identified [68]. The classical model proposes a hierarchical structure comprising distinct stem and progenitor cells [69,70]. Keratinocyte stem cells are capable of nearly unlimited self-renewal, though they typically exist in a quiescent state and rarely divide [71]. Transit-amplifying cells, which originate from stem cells, demonstrate high proliferative capacity and multipotency, undergoing multiple rounds of cell division prior to differentiation [68,72]. Progenitor cells migrate to the skin’s surface to replace terminally differentiated dead corneocytes [66]. This intricate process entails the spatial and temporal control of a multitude of genes. While the regulatory mechanisms remain poorly understood, a substantial body of evidence suggests that balanced epigenetic regulation plays a pivotal role [73].

DNA modifications have been identified as playing a critical role in the regulation of epidermal differentiation [67]. This is substantiated by the existence of a DNA methylation gradient between the various skin layers, as well as the distinctive pattern of genes involved in de novo and maintenance methylation. Dnmt1 is essential for the self-renewal of progenitor cells in diverse somatic tissues, thereby maintaining the distinctive DNA methylation pattern characteristic of stem cells. This function is consistent with the observation that Dnmt1 is exclusively observed in the basal layer of the epidermis, where it plays an important role in maintaining the progenitor cell status of this zone [74]. The establishment and maintenance of tissue identity are achieved through the methylation-dependent repression of non-skin genes and the activation of specific genes required for the epidermal lineage during the early stages of differentiation [75,76]. For the continuous self-renewal of the basal stem cells, it is also essential that they retain their proliferative capacity and repress differentiation processes, in which Dnmt1 plays a pivotal role. Furthermore, evidence indicates that DNMT1 deficiency results in the loss of stem cell characteristics, which, in turn, leads to premature differentiation [74]. In addition, Dnmt1, which is responsible for maintaining DNA methylation, and Dnmt3A and Dnmt3B, which are responsible for de novo methylation, also exhibit high expression levels in the basal layer [77].

Upon exiting the stratum basale, keratinocytes cease proliferation and initiate their differentiation program [78,79]. Concurrently, the amount of Dnmt3B declines as cells advance to the suprabasal layers [76]. In this phase, active DNA demethylation (regulated by, for example, GADD45) also plays a role in the differentiation of keratinocytes [74]. The regulation of specific gene complexes is a hallmark of these processes. These include keratins, which are located at two different chromosomal complexes, 17q21 and 12q13 [80], and the epidermal differentiation complex (EDC) on chromosome 1q21. This latter region contains over 50 genes, including the late cornified envelope family, the S100 family, the S100-fused type, and the small proline-rich protein families [81]. These factors regulate and maintain the formation of the stratified epithelium and cornified envelope [82]. It is noteworthy that while the overall transcription rate gradually decreases during epidermal differentiation [83], EDC gene expression exhibits an opposite trend, indicative of highly specific and coordinated regulatory elements governing the expression of these genes [84].

The precise regulation of chromatin’s structure represents an additional means of gene expression regulation in the adult epithelium [48]. Keratinocyte stem cells exhibit elevated levels of H3K9me3 and H4K20me3, accompanied by a reduction in histone H4 acetylation and H4K20me14. As quiescent stem cells undergo a transformation into transit-amplifying cells, there is an observed increase in the level of histone H4 acetylation while the level of histone H3K9 methylation decreases [49]. Concurrently, with these alterations, transit-amplifying cells manifest a markedly proliferative phenotype, undergoing multiple rounds of division before reaching the suprabasal layer and commencing differentiation [49,50]. A typical histone epigenetic gradient is also observed between the epidermal layers, with high levels of H3K4me3, H3K27ac, and H4K16ac and low levels of H3K27me3 and H4K20me3 in the basal layer. In contrast, the suprabasal layer is distinguished by elevated levels of H3K27me3, which is an expansion of H4K20me3 heterochromatic foci and a reduction in H4K16ac [51].

ncRNAs also play a significant role in these processes. The importance of miRNAs is substantiated by the observation that the disruption of specific members of the miRNA regulatory system (Dicer, Dgcr8) and the RISC complex (Ago1, Ago2) in knock-out mice results in a profound disruption of cutaneous homeostasis [63,67,85,86]. In recent years, research has also demonstrated the role of several miRNAs in the maintenance and differentiation of stem cells, as well as in the regulation of epidermal homeostasis and the proper balance of epidermal proliferation and differentiation [67], in which the p63 transcription factor is a key regulator. Several miRNAs play a role in the regulation of p63, and it also regulates a number of other miRNAs. This complex system is important for the cell cycle exit of basal keratinocytes and also later in regulating the balance of proliferation and differentiation processes [83].

In recent years, a number of lncRNAs have been shown to play significant regulatory roles, largely due to their capacity to bind to DNA, RNA, and protein molecules. Accordingly, these properties enable them to affect a wide array of molecular processes. For instance, they can function as transcription regulators. Specific ncRNAs have the capacity to recruit histone-modifying complexes to particular genomic loci, resulting in alterations in chromatin states that either facilitate or inhibit the nearby loci. Additionally, they may interfere with mRNA activity and processing, regulate miRNAs, and modify protein location and function [87]. It is, therefore, unsurprising that they act as pivotal regulators of the balance between stem cell self-renewal and epidermal differentiation. Notable examples include DANCR (differentiation antagonizing non-protein coding RNA) [88] and TINCR (terminal differentiation-induced ncRNA) [89]. These lncRNAs play crucial roles in these processes. While DANCR is necessary for the maintenance of the undifferentiated state, TINCR is a crucial important regulator of keratinocyte differentiation [90]. BLNCR is another pivotal factor in these processes, acting as a central molecule for epidermal stem cell maintenance [91]. During the process of skin development, lncRNAs play a crucial role in modulating the expression of keratins and other differentiation-related molecules. These include the molecules involved in the formation of the cornified envelope and peptide cross-linking, which are essential for maintaining the structural integrity and optimal functioning of the epidermis. One important regulator of these factors is LINC00041, which may serve a role in fine-tuning keratinocyte differentiation by antagonizing the differentiation-promoting SPRR5 protein [90,92]. It is evident that our comprehension of the function of ncRNAs in skin development is still incomplete. The involvement of lncRNAs in interactions with microRNAs (miRNAs) alongside small interfering RNAs (siRNAs) and potentially other types of ncNRAs adds another layer of complexity to the overall picture, as it allows for the precise regulation of gene expression involved in skin development.

### 3.2. Skin Aging

Aging is the natural time-related deterioration of an organism that increases the probability of death [93]. Although approximately 20–30% of this process is genetically determined, the majority is thought to be determined by epigenetic changes [94]. The process of skin aging is a complex biological phenomenon, the specifics of which remain incompletely understood. Phenotypically, aging is manifested by a thinning of the skin and hair, increased fragility, and the formation of wrinkles. Concomitantly, the accumulation of cellular damage ultimately contributes to a gradual decline in skin functions, decreased immunity and barrier functions, and the frequent accumulation of senescent cells due to their incomplete clearance. This results in the appearance of age-related diseases, infections, and skin cancers [95].

The process of aging is influenced by a combination of intrinsic and extrinsic environmental factors. While the intrinsic factors determine an individual’s genetic predisposition to aging, extrinsic factors such as sun exposure, pollution, and lifestyle choices can accelerate this process [96,97,98]. The field of epigenetics has emerged as a pivotal area of research in understanding the molecular mechanisms of these processes. Changes in DNA and RNA methylations, histone modifications, and non-coding RNA expression have emerged as pivotal regulators [99,100,101].

One of the most extensively studied epigenetic modifications in the context of skin aging is DNA methylation. Studies have demonstrated that its pattern is tissue-specific and that age-related changes occur in a time-dependent manner in keratinocytes and fibroblasts [102]. These changes have been demonstrated to affect major keratinocyte functions, including the hypermethylation of promoter regions of genes involved in processes such as keratinocyte differentiation and barrier function [103]. Furthermore, reports indicate a reduction in collagen production, an increase in inflammation, and impaired wound healing [104].

A significant phenomenon associated with the natural aging process is the thinning of the skin, which is most evident in the epidermal compartment. Although the number of stem cells in the epidermis appears to remain relatively stable with age, the capacity to produce differentiated cells is diminished due to a slower cell cycle compared to that observed in younger cells [105,106]. The p16INK4a protein, a cyclin-dependent kinase inhibitor, plays a significant role in these processes [106,107,108]. Its expression is observed to increase gradually with age due to the hypermethylation of its promoter region in aged skin [109]. Another crucial element is the p63 protein, which plays a pivotal role in maintaining epithelial integrity, regulating skin development, and facilitating regeneration. It exists in multiple isoforms with different functions, and studies have demonstrated that the ΔNp63 isoform is primarily involved in maintaining stem cells. The levels of ΔNp63 are reduced in aged skin, which correlates with a diminished capacity for regeneration and impaired wound healing [110] (Figure 5).

Methylation changes also affect fibroblasts. In this cell type, the altered chromatin structure in fibroblasts from old donors was first reported in the 1980s. This suggested that these changes play a significant role in their age-related functional impairment [111]. Subsequently, it was revealed that the expression of the DNMT1 maintenance methylase is also diminished in aged fibroblasts. This results in a loss of methylation stability, which, in turn, leads to an increased incidence of senescence in human skin [112,113]. Furthermore, aged skin exhibits a reduction in the number of cells synthesizing collagen [114]. The resulting changes have a marked impact on extracellular matrix (ECM) formation and maintenance, which, in turn, give rise to many of the aforementioned phenotypic changes observed in aged skin. In addition, genes linked to collagen synthesis and remodeling display altered methylation status, resulting in reduced expression and contributing to the loss of skin elasticity and firmness [115]. One such example is the COL1A1 gene, which has been observed to exhibit diminished expression with age [116]. Concomitantly, elevated levels of collagen-degrading enzymes, specifically matrix metalloproteinases (MMPs), are also produced by the cells. The altered equilibrium between ECM production and degradation gives rise to a vicious cycle that further accelerates the aging process [117].

Tissue-specific “epigenetic clocks” are now available, which utilize DNA methylation patterns to predict biological age. These employ specific methylation markers and are capable of accurately reflecting the chronological age of an individual’s skin, as well as predicting age-related changes in skin appearance [118,119].

Another pivotal epigenetic mechanism implicated in cutaneous aging is histone modification. A significant aspect of these events is the alteration of genomic organization, which is influenced by changes in histone composition [120]. In aged skin, the histone variant H2A.J has been detected in both the keratinocyte and fibroblast compartments, which may correlate with increased cellular senescence, alterations in chromatin properties, and the increased expression of inflammatory mediators [121,122]. In addition to alterations in DNA methylation, the histone acetylation pattern also undergoes changes with age. It is of great importance to maintain the acetylation pattern characteristic of younger cells, as changes in histone acetylation and deacetylation events are frequently observed in malignant processes. As the expression of numerous HDACs (histone deacetylases) is elevated in a multitude of cancers, their inhibitors may prove to be a valuable therapeutic option [123]. With regard to the process of aging, these agents appear to exhibit potential anti-aging effects by promoting the expression of genes involved in the repair and regeneration of the skin [124,125].

Non-coding RNAs, including microRNAs (miRNAs) and long non-coding RNAs (lncRNAs), also play a significant role in regulating gene expression and contributing to the process of skin aging. A number of miRNAs have been identified as being differentially expressed in young versus elderly individuals, and some of these even serve as biomarkers (Table 1). These regulate the principal biological processes associated with the aging process, including cellular senescence, inflammation, and extracellular matrix remodeling. The identification of the specific targets regulated by these ncRNAs may provide insight into the role of currently unknown aging-related mechanisms and cellular processes [126].

Significant alterations occur during the aging process of tissues, affecting the telomeres, which are repetitive sequences located at the end of each chromosome. Telomere length diminishes with age, and the progressive shortening of telomeres has been demonstrated to precipitate cellular senescence, which significantly impacts the health and lifespan of the tissue and the organism as a whole [127]. A correlation has been established between shorter telomeres and an elevated risk of disease, as well as a reduced survival rate. Telomere dysfunction is closely associated with replicative senescence, which affects numerous cellular functions, including chromatin organization, metabolism, and gene expression [93]. During senescence, an intriguing phenomenon is the emergence of a pro-inflammatory phenotype, also termed the senescence-associated secretory phenotype (SASP). This intricate process encompasses metabolic alterations, including mitochondrial reactive oxygen species (ROS) overproduction, the cell-type-specific activation of innate immune signaling pathways, and disturbances in autophagic processes. These factors collectively contribute to tissue inflammation and the progression of aging.

**Table 1 ijms-26-00438-t001:** List of the best-known cutaneous aging-related ncRNA biomarkers.

ncRNA Biomarker	Type	Major Function Related to Aging	References
let-7 family	miRNA	longevity-related	[125]
miR-34 family	miRNA	regulation of senescence and apoptosis	[126,128]
miR-130 family	miRNA	regulation of keratinocyte proliferation and differentiation (senescence)	[126]
miR-146a	miRNA	anti-inflammatory and promotes survival under stress conditions	[126,129]
miR-181family	miRNA	regulation of immunosenescence and mitochondrial function	[126,130]
miR-221/222	miRNA	regulation of proliferation and survival pathways	[126,131]
H19	lncRNA	regulation of senescence	[132]
PANDA	lncRNA	regulation of p21 expression, which is implicated in DNA damage response	[133,134]
SPRR2C	lncRNA	modulating calcium signaling pathways, influencing cutaneous differentiation	[132,135]
GAS5	lncRNA	regulation of growth arrest and apoptosis, influencing cellular responses to stress	[136]
TERRA	lncRNA	telomere maintenance and DNA damage response	[137]

The extrinsic factor that exerts the most profound impact on the process of skin aging is ultraviolet (UV) radiation emitted from the sun [138]. It has been demonstrated that UV radiation induces DNA damage, oxidative stress, and inflammation, all of which contribute to epigenetic changes and accelerate the aging process [87]. Additionally, UV-induced DNA damage can also result in aberrant DNA methylation patterns, while oxidative stress can promote histone modifications and alter miRNA expression [87]. Furthermore, chronic exposure to UV radiation can also induce cellular senescence [139].

In addition to UV radiation, other environmental factors, including pollution, smoking, and poor diet, have also been demonstrated to contribute to epigenetic changes and accelerate the aging of the skin [140,141,142,143]. These factors have the potential to induce oxidative stress, inflammation, and DNA damage, which may result in alterations to DNA methylation, histone modifications, and miRNA expression. For example, studies have demonstrated that exposure to air pollution is associated with the increased DNA methylation of specific genes involved in the skin barrier function [143,144].

### 3.3. Epigenetics and Microbiota

The skin, like other barrier organs, is home to a vast array of microbes, including bacteria, viruses, fungi, and archaea. These microbes have evolved to flourish in our environment and have adapted to our conditions to a considerable extent over the course of evolution. This microbial community is referred to as the microbiota [145]. Given their abundance and intimate relationship with the cells in our body, it is unsurprising that they play an essential role in maintaining the healthy structure and function of our organism [146]. The existing research indicates that members of microbiota may also influence the function of human cells through direct physical contact or through their metabolites, affecting cellular signaling and epigenetic regulatory processes. Moreover, the intricacy of this system is augmented by the fact that, in a specific tissue, not only are the constituents of the microbial community directly associated with it but also microbes that colonize distant organs may exert their effects via the bloodstream.

However, the majority of what is known in this regard has been derived from studies of the gut microbiota [147,148,149,150]. In light of these findings, a current hypothesis posits that microbes may influence the expression and activity of enzymes involved in epigenetic regulation in human cells. Additionally, microbial metabolites may serve as donors for DNA or histone modifications, and they may influence the efficiency of certain epigenetic regulatory processes [151].

Individual members of microbiota are capable of synthesizing and secreting metabolites that are relevant to epigenetics, which can modulate the host gene expression. The most promising of these molecules are short-chain fatty acids (SCFAs), including acetic acid (C2, AA), propionic acid (C3, PA), butyric acid (C4, BA), and valeric acid (C5, VA), which are produced by various microbes [147,151,152,153]. The results of animal experiments indicate that a reduction in the number of SCFAs present in the alimentary canal, for instance, in the absence of gut microbiota under germ-free conditions or in mice maintained on a special diet under normal conditions, may have a significant impact on the pattern of histone modifications in the genome. This suggests that microbiota-produced SCFAs play a pivotal role in epigenetic regulation [154]. One reason for their genomic effect is that these molecules, in particular PA, BA, and VA, have been demonstrated to interfere with HDAC activity, resulting in increased histone acetylation [155,156].

While the primary source of these compounds is the gut, where anaerobic fermenting microbes produce them in large quantities, some members of skin-colonizing microbes also possess similar properties. *Cutibacterium acnes* (*C. acnes*) and *Staphylococcus epidermidis* (*S. epidermidis*), which are ubiquitous members of the cutaneous microbiota, serve as prominent examples [157,158]. The question of whether these microbes exert differential effects on epigenetic regulatory processes remains to be elucidated through systematic, comparative studies, which will be a subject of future research.

In vitro results clearly indicate that SCFAs produced by *C. acnes* are capable of inhibiting the activity of HDAC 8 and 9, thereby influencing the outcome of TLR activation and subsequent immune and inflammatory processes in keratinocytes and sebocytes [147,159]. Given these functions, it is evident that qualitative and/or quantitative alterations in the composition of the cutaneous microbiota exert a profound impact on the epigenetic regulation of human cells. The resulting metabolomic variations in the resident microbial community due to differences in species composition can be considered a significant epigenetic factor [160].

The occurrence of epigenetic modifications, including alterations in DNA and histone methylation and acetylation, is contingent upon the availability of methyl and acetyl donors within the cells. While the enzymes that catalyze these changes primarily utilize the sources generated by cells themselves during cellular metabolic pathways, in certain instances, the gut microbiota may also serve as a source [161]. The question of whether this pathway is also utilized in the skin remains to be answered, but based on the similarities regarding the functions and properties of these two microbial communities, it seems a reasonable assumption.

In recent years, significant advances have been made in our understanding of how members of the microbiota impact the molecular environment of the skin (Figure 6). The most well-studied members, *C. acnes* and *S. epidermidis* have been shown to influence the skin’s molecular environment through immune and inflammatory activation and barrier properties, respectively [162,163,164]. This shift can result in the production of pro-inflammatory cytokines, including IL-1β, TNF-α, and IL-6, through the activation of nuclear factor kappa B (NF-κB), mitogen-activated protein kinase (MAPK) signaling, and inflammasome pathways [165]. These responses indirectly influence the expression of several miRNAs, and among these, mir-146 has been the subject of the most extensive study. *C. acnes* biofilms have been demonstrated to directly affect the expression of mir-146 through Toll-like receptor 2 (TLR2)-mediated signaling events in keratinocytes [166]. Similarly, analogous pathways are operational in sebocytes, and in both cell types, miR-146 exerts a negative regulatory function on immune effector expression [167].

Cutaneous microbes appear to be capable of modulating each other’s effects, as evidenced by recent research. For example, the LTA of *S. epidermidis* has been shown to play a crucial role in modulating *C. acnes*-induced inflammatory responses through the induction of miR-143 in keratinocytes [168]. These effects may be of critical importance for maintaining skin homeostasis, potentially contributing to the tolerance of microbes in the skin.

An imbalance in microbial communities, or dysbiosis, may also have significant effects on the cellular level. Such occurrences may be linked to increased inflammation, which can be attributed to the influence of cytokines and other inflammatory mediators [169,170]. An example includes acne vulgaris, in which dysbiosis involving *C. acnes* is a significant contributing factor [171]. A number of miRNAs (e.g., mirR-21, miR-150, and miR-223) have been linked to the pathogenesis of this common skin disease [172]. However, further research is needed to determine the direct impact of microbiota on the regulation of these miRNAs.

It seems that such alterations leave molecular traces in the affected cells. It has been demonstrated that imiquimod-induced inflammatory events in mouse skin result in epigenetic changes in epidermal stem cells, leading to differential responsiveness following repeated activation, which is caused by cellular reprogramming [173]. This particular keratinocyte memory is the result of innate immune memory processes initially described by Netea and colleagues in immune cells [174,175]. Epigenetic regulation, particularly involving DNA methylation and histone acetylation processes, appears to play an important role in their development [176,177]. Although our current understanding of their precise role and molecular characteristics remains limited, these processes may play a significant role in the recurrence of lesions in the same skin areas in chronic inflammatory diseases, as observed in psoriatic patients [178].

As previously outlined, substantial structural and functional alterations occur in the skin as it ages, accompanied by distinctive shifts in the cutaneous microbiota. A notable alteration observed across geographically disparate populations is the pronounced decline in the relative abundance of *C. acnes* at all sampled body sites [179,180,181]. A potential reduction in the quantity of SCFAs resulting from these changes may also occur, which could have a considerable impact on epigenetic regulatory processes in human cells. Furthermore, aging is accompanied by alterations in immune cell composition and a decline in cutaneous immunity, which collectively result in compromised barrier functions [182,183].

## 4. Discussion

The complex interplay of intrinsic (genetic, epigenetic) and extrinsic factors (environmental changes, diet, smoking, UV exposure) plays a pivotal role in the maintenance of homeostatic processes in the human body. Disruptions in this intricate system can lead to pathophysiological processes. A mounting body of evidence, both direct and indirect, suggests that the microbiota play a complex and important role. A thorough examination of the constituents of this intricate network, their interactions, and the impact of diverse alterations on the system’s overall functionality may provide a foundation for future research endeavors.

A mounting body of research has revealed the significant impact of the most prevalent bacterial species in the skin microbiota on our ecosystem. At present, the role of different fungal and viral species is much less well understood. However, these species may also be able to activate various pattern recognition receptors and, thus, directly influence human cells [184,185]. Furthermore, these organisms may also affect the composition and functioning of the bacterial community through population interactions [185,186]. An example includes the bacteriophages, which are abundant and likely play a role in regulating the amount and composition of bacterial populations. This, in turn, may also exhibit an indirect effect on the skin [187].

The findings presented in this manuscript underscore the critical role of epigenetic mechanisms in maintaining skin homeostasis and facilitating adaptive responses to environmental changes. As highlighted, the skin is not only a barrier but also a dynamic organ that undergoes continuous renewal and differentiation, which are processes intricately regulated by epigenetic modifications. The notion that epigenetic regulation allows for rapid adaptation to environmental stimuli is particularly relevant in the context of skin physiology, with the skin being at the interface between our body and the external environment. Given the rapid pace of environmental changes since the Industrial Revolution, understanding how these factors influence epigenetic regulation is paramount.

The interplay between epigenetics and skin pathologies such as psoriasis, eczema, and skin cancers suggests that the dysregulation of epigenetic mechanisms may lead to the pathogenesis of these diseases, involving chronic inflammatory responses or malignant transformations. Understanding these connections may open avenues for developing novel therapeutic strategies that target epigenetic regulators to restore normal gene expression patterns in diseased skin.

Non-coding RNAs represent a burgeoning area of interest in epigenetic regulation. Their involvement in modulating chromatin’s structure and influencing gene expression adds another layer of complexity to our understanding of skin biology. They may also serve as biomarkers and/or therapeutic targets for skin diseases.

Based on the evidence provided, it is advisable to advance our understanding of epigenetic regulation in skin biology, and future studies should employ an integrative approach to combine these with genomic, transcriptomic, and other omic data.

## Figures and Tables

**Figure 1 ijms-26-00438-f001:**
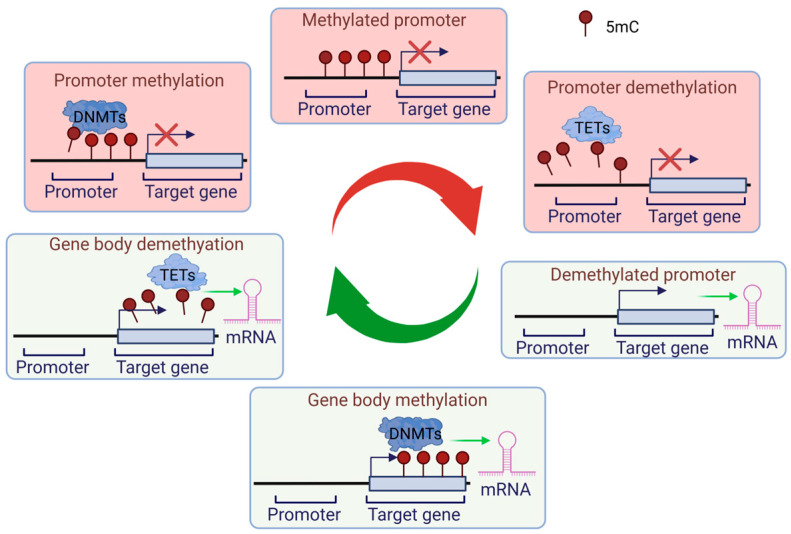
DNA methylation-mediated epigenetic regulation of gene expression.

**Figure 2 ijms-26-00438-f002:**
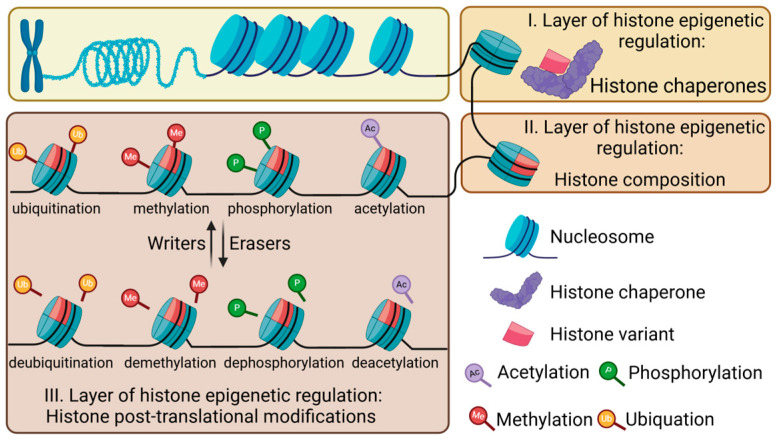
The three major layers of histone-related epigenetic regulation.

**Figure 3 ijms-26-00438-f003:**
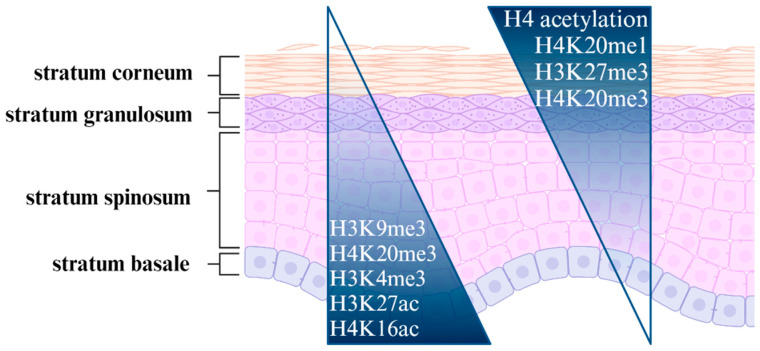
Histone modifications exhibit a distinctive spatial pattern across the various layers of the epidermis, which plays a pivotal role in regulating keratinocyte differentiation processes and, subsequently, the formation of a stratified epidermis [48,49,50,51].

**Figure 4 ijms-26-00438-f004:**
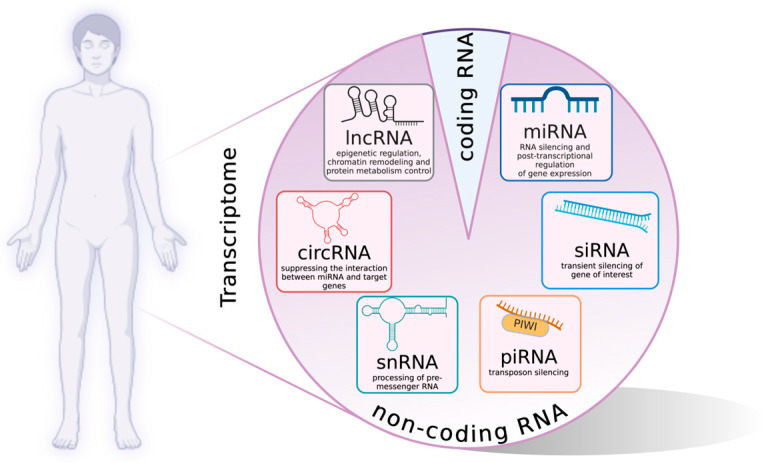
The human genome contains a significant predominance of non-coding regions, which constitute approximately 98% of the genome’s total length. The remaining portion is composed primarily of non-coding RNAs, including microRNAs (miRNAs), small interfering RNAs (siRNAs), iwi-interacting RNAs (piRNAs), small nucleolar RNAs (snoRNAs), and long non-coding RNAs (lncRNAs). With regards to skin, the function of specific miRNAs and lncRNAs has been most extensively investigated (for further details, please refer to the corresponding text).

**Figure 5 ijms-26-00438-f005:**
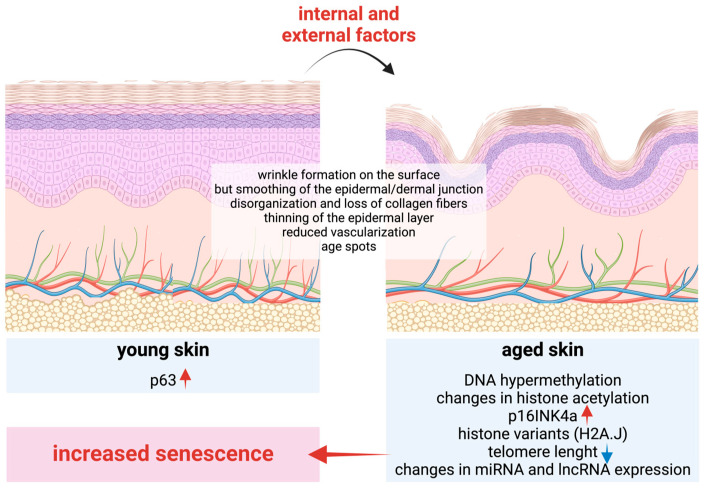
Complex molecular changes occur during the skin’s aging process, resulting in significant structural changes. (Red arrows indicate increased, while blue ones decreased values.)

**Figure 6 ijms-26-00438-f006:**
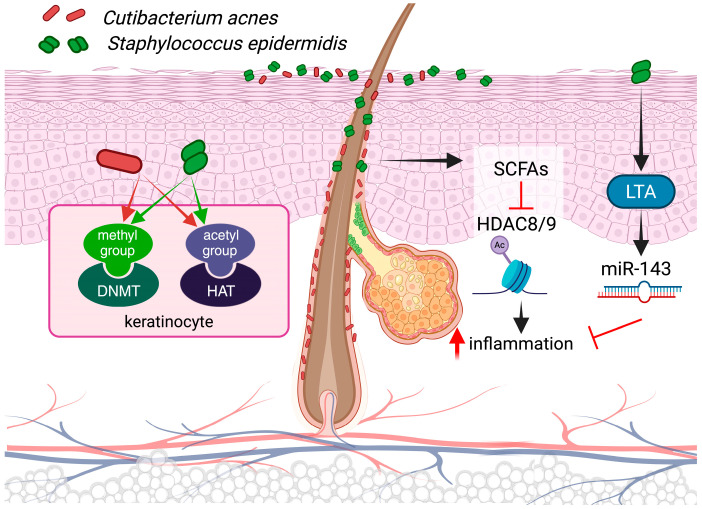
The microbiota may exert intricate influences on the epigenetic regulation of keratinocytes. The activation of pattern recognition receptors may trigger signaling processes that directly influence the expression of various non-coding RNAs. The SCFA molecules they produce can serve as methyl and acetyl donors and also have HDAC inhibitory activity. Furthermore, the influence of individual species on each other may be subject to complex interactions, particularly in the context of dysbiotic processes.

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
