# Peer review of "Epigenetic Regulatory Processes Involved in the Establishment and Maintenance of Skin Homeostasis—The Role of Microbiota"

_ijms, 2025, doi:10.3390/ijms26020438_

Round 1

Reviewer 1 Report

Comments and Suggestions for Authors

This is a well-written comprehensive review of epiregulatory mechanisms in the skin.

The manuscript may benefit from some final polishing to be fully acceptable.

For example, although very attractive, Figur4 does not show any functional/mechanisms of the different types of RNA.

In the section Skin development, for example, there is an overuse of the word "pivotal" (lines 216-219, and then 228

Section 3.2 on aging is so long that it would benefit from the addition of a diagram. Same for 3.3

Author Response

We would like to express our gratitude to the reviewer for their assessment of our manuscript and for their encouraging feedback.

  1. This is a well-written comprehensive review of epiregulatory mechanisms in the skin. The manuscript may benefit from some final polishing to be fully acceptable.For example, although very attractive, Figur4 does not show any functional/mechanisms of the different types of RNA.

As recommended, modifications have been made to Fig. 4. Please refer to the manuscript for the revised version.

  1. In the section Skin development, for example, there is an overuse of the word "pivotal" (lines 216-219, and then 228

We are grateful for this observation, and we have rephrased the aforementioned sections.

  1. Section 3.2 on aging is so long that it would benefit from the addition of a diagram. Same for 3.3

As suggested, we have designed two additional figures. Please refer to the new version of the manuscript.

Reviewer 2 Report

Comments and Suggestions for Authors

In this review, the authors discussed the role of the microbiota in epigenetic regulatory processes involved in the establishment and maintenance of skin homeostasis. The authors discussed the following points: Epigentic regulation & Epigentic regulation of histones, histones cheaprones, and posttranslational modifications of hostones, Crosstalk among different epigenetic regulatory mechanisms. Besides Epigentic regualtion among healthy and aging skin

Unfortuntaley the idea is not novel, and proviously discussed in several reports such as

https://pmc.ncbi.nlm.nih.gov/articles/PMC8359218/

https://cancerci.biomedcentral.com/articles/10.1186/s12935-022-02738-0

So what are new directions of this review?

Besides, please consider the followings

a) What are the most common microbiota including in epigenetic regulation of skin? Difference between each microbe?

b) Source of these microbiota that affect the skin heamostasis

c) How interaction between genetic, epigentic, enviormental and microbiota affect skin heamostatsis

d) Which microbiota affect mainly: bacteria, mycobiome, or virome.

Author Response

  1. In this review, the authors discussed the role of the microbiota in epigenetic regulatory processes involved in the establishment and maintenance of skin homeostasis. The authors discussed the following points: Epigentic regulation & Epigentic regulation of histones, histones cheaprones, and posttranslational modifications of hostones, Crosstalk among different epigenetic regulatory mechanisms. Besides Epigentic regualtion among healthy and aging skin

Unfortuntaley the idea is not novel, and proviously discussed in several reports such as

https://pmc.ncbi.nlm.nih.gov/articles/PMC8359218/

https://cancerci.biomedcentral.com/articles/10.1186/s12935-022-02738-0

So what are new directions of this review?

We are grateful to the reviewer for bringing these articles to our attention. We agree, that in recent years, an increasing number of publications have emerged that describe the role of specific elements of epigenetic regulation in various cellular biological processes of the skin. However, the aforementioned studies have a primary focus on the epigenetic regulation of skin diseases, including melanoma, psoriasis, and atopic dermatitis.

The article by Wagner et al. that was referenced by the reviewer does indeed describe some epigenetic regulatory processes involved in establishing the homeostatic conditions of the skin. However, it does so in much less detail than our own review. Furthermore, the article has a distinct primary focus, namely epigenetic regulation in epidermal stem cells and the pathogenesis of epidermolysis bullosa.

In comparison, our review focuses on epigenetic regulation during skin homeostasis and provides a comprehensive overview of the currently known elements at all levels of the epigenetic regulatory system in terms of skin development and aging.

The other article by Fath et al. focuses on the epigenetic regulation in melanoma development.

We believe that currently the role of the microbiota in host epigenetic regulation is currently less well-understood in the context of the skin. Accordingly, we have summarized the available literature data and identified the potential involvement of the cutaneous microbiota in these processes.

  1. Besides, please consider the followings
  2. a) What are the most common microbiota including in epigenetic regulation of skin? Difference between each microbe?

The majority of available data regarding the impact of the skin microbiota on skin cells is concentrated on the commensal Cutibacterium acnes (C.            acnes) and Staphylococcus epidermidis (S. epidermidis), in addition to the pathogenic Staphylococcus aureus (S. aureus). C. acnes and S. epidermidis are ubiquitous on the human skin and are thought to play a role in regulating the host epigenetic processes, both directly and through their metabolites. Given the paucity of data concerning their precise roles in these processes, it is currently unclear whether differences exist in their effect on these processes. To address these questions, it would be necessary to conduct systematic, comparative studies, which could form the subject of future research.

  1. b) Source of these microbiota that affect the skin heamostasis

It is possible that primarily the members of the skin microbiota that, through direct physical contact or through their metabolites, can influence the function of skin cells and the epigenetic regulatory processes that occur within them. Furthermore, microbes that colonize distant organs, including metabolites of microbes in the alimentary canal, may also exert their effects in the skin via the blood.

  1. c) How interaction between genetic, epigentic, enviormental and microbiota affect skin heamostatsis

We believe that the intricate interplay of intrinsic (genetic, epigenetic) and extrinsic factors (environmental changes, diet, smoking, UV exposure) plays a pivotal role in the maintenance of homeostasis. When this intricate system is not in equilibrium, it may also instigate pathogenic processes. An increasing body of evidence, both direct and indirect, suggests that the microbiota plays a complex and important role. A comprehensive examination of the constituents of this complex network, their interactions, and the impact of diverse alterations on the system's overall functionality may provide a foundation for future research endeavors.

  1. d) Which microbiota affect mainly: bacteria, mycobiome, or virome.

There is limited data on how the microbiota (bacteria) influences epigenetic regulation in the skin and, consequently, how they contribute to the establishment of homeostatic conditions of healthy skin, as well as in disease. However, it appears that they contribute both directly and indirectly through their metabolites. Regarding the skin mycobiome, most of the data is focused on Malassezia species found in moist skin areas; however, to our knowledge, there is no information regarding their potential role in skin epigenetic regulation. Regarding the skin virome, there is no direct data on its involvement in epigenetic regulation. However, we believe that certain bacteriophages, for example, may indirectly contribute through the bacteria.

Round 2

Reviewer 2 Report

Comments and Suggestions for Authors

Although the authors replied incompletely to my questions, none of the replies were added to the revised manuscript.

Therefore, I would suggest the authors to include the replies in the revised manuscript.

Author Response

12/28/2024.

Answers to Reviewer 2.

We would like to thank the anonymous reviewer for taking the time to review our manuscript and for his comments. Based on these, elements of our responses to the questions posed in the first round of the review process have been incorporated into the text. Our changes are highlighted in yellow in the current version of the paper.

Sincerely,

Kornélia Szabó

(corresponding author)

Round 3

Reviewer 2 Report

Comments and Suggestions for Authors

No further comments